# Effects of Boron Content on the Microstructure and Impact Toughness of 12Cr1MoVR Low-Alloy Heat-Resistant Steel Weld Metals

**DOI:** 10.3390/ma14040926

**Published:** 2021-02-15

**Authors:** Guishan Dou, Rui Cao, Changliang Cai, Cheng Han, Xili Guo, Yong Jiang, Jianhong Chen

**Affiliations:** 1State Key Laboratory of Advanced Processing and Recycling of Non-Ferrous Metal, Lanzhou University of Technology, Lanzhou 730050, China; douguishan2021@163.com (G.D.); tuopang441@163.com (C.C.); chenghan0126@163.com (C.H.); zchen@lut.cn (J.C.); 2Department of Materials Science and Engineering, Lanzhou University of Technology, Lanzhou 730050, China; 3Atlantic China Welding Consumables, Inc., Zigong 643000, China; xiaoxi2002_1@163.com (X.G.); jy70c6@yeah.net (Y.J.)

**Keywords:** low-Cr heat-resistant steel weld metal, impact toughness, microstructure, hardness, boron

## Abstract

The impact toughness of low-Cr heat-resistant steel weld metal is an important problem to broaden the application of low-Cr heat-resistant steel. In this study, the microstructure and impact toughness of 12Cr1MoVR low-alloy heat-resistant steel weld metals with various boron contents (B1 = 0.0028%, B2 = 0.0054%, and B3 = 0.0079%) were investigated. The microstructures of all weld metals were composed of block ferrite, carbides, and inclusions. Results indicated that with increased B content, prior austenite grain sizes decreased, and minor microstructure changes could be found. With the increase in B content from 0.0028% to 0.0054% to 0.0079%, the ductile–brittle transition temperature of the weld metals decreased from 30 to 0 to −14 °C, the toughness of weld metal increased, and the hardness slightly decreased, all of which are directly related to the refinement of prior austenite grain size because of the addition of B content. However, on the top-shelf zone, such as at the testing temperature of 80 °C, ductile fracture dominates the fracture surface; with the increase in B content, the size and density of inclusions increased gradually, which led to the decrease of the impact toughness at 80 °C when the B content was 0.0079%.

## 1. Introduction

In recent years, the performance field of the Cr–Mo heat-resistant steel has been used in high temperature and pressure applications that include critical components of modern supercritical and ultra-supercritical thermal power plants. Because the applications become more and more extensive, additional investigations concentrated on the development of heat-resistant steels with harsh service conditions including high temperatures (from 750 °C up to 1100 °C) are needed. In order to promote the application of low-Cr heat-resistant steel, it is important to obtain the welding joints with enough heat resistance and impact toughness. The impact toughness of the welding joints is directly related to the microstructure of the welding joints. The microstructure of the welding joints is directly related to the composition and the welding process. The microstructure of steel includes ferrite, pearlite, bainite, and martensite. The microstructures with the best toughness are acicular ferrite, lower bainite, and low-carbon lath martensite. In addition to the microstructure types, another key factor affecting the impact toughness is grain size.

The main components of heat-resistant steel are Cr and Mo. The research on Cr and Mo has been greater and clearer. Cr can promote the formation of carbides in heat-resistant steel, improving the high-temperature creep strength [1,2]. At the same time, it can improve the hardenability and strength of weld metals [3,4]. However, too high of a Cr content induces the formation of side-plate ferrite containing a Martensite-Austenite (M-A) constituent, which decreases the toughness of the weld metals [4]. The addition of Cr can improve the oxidation resistance and corrosion resistance of the weld metals [3]. Mo can effectively inhibit the formation of cementite and can promote the precipitation of carbide, so it can significantly improve the creep strength and rupture strength of heat-resistant steel [1]. Mo can produce solute drag effects on ferrite and bainite transformations [5,6]. Mo provides a significant increase in hardness of ~20% compared to the base material without Mo [6]. S. D. Bhole [7] has shown that the increase in Mo can inhibit the formation of grain boundary ferrite (GBF) and can increase the proportion of acicular ferrite, which finally improves the toughness of the weld metal. Trace boron can significantly improve the hardenability of steel and can save resources by replacing Cr, Mo, Ni, and other expensive metals.

Other researchers have investigated the effects of B on impact toughness. The influence of boron on the impact toughness of boron-containing steel mainly depends on the boron content and the synergistic effect of B and Ti. A lower amount of B can increase toughness because of the refinement of the grain size [8]. Excessive B leads to the formation of M-A constituents and leads to a decrease in the acicular ferrite, which results in the impact toughness decreasing [9]. The effects of B on the microstructure and impact toughness are directly related to other alloy elements. Ti can protect B and can combine into TiB_2_. The effect of different boron contents on the impact toughness of boron-containing steel is different. Zhao et al. [8] found that the uneven distribution of boron suppressed the formation of GBF and promoted the formation of intragranular ferrite (IGF) idiomorphs, which increased the impact toughness. Hu et al. [10] showed that the IGF nucleation on particles in austenite refined the microstructure of steels. Rodríguez-Galeano et al. [11] indicated that with the increase in B content from 0 ppm to 60 ppm, the grain size of hard phases, such as the martensite phase or M-A constituents, decreased, which induced higher toughness. Klimenkov et al. [12] reported that B content also affected the density and composition of M_23_C_6_ and MX precipitates, the appearance of BN, and affected the microstructure. Astini et al. [13] found that the addition of 0.04% boron to the High Strength Steel (HSS) alloys increased the bending strength by more than 10% because the addition of boron reduced the cell size and produced finer carbide. Zhu et al. [14] indicated that the addition of B only slightly decreased the bainite transformation temperature at low cooling rates (≤10 °C/s), whereas the combined addition of B + Nb greatly decreased the transformation temperature. Shi et al. [15] found that the addition of B made the ferrite transformation temperature increase from 775 to 800 °C, the amount of polygonal ferrites increase from 63.2% to 78.6% and the ductile-to-brittle transition temperature decreases from −30 to −50 °C. Additionally, they found that the sizes of (Ti,V)(C,N)-MnS-BN, MnS-BN, Al_2_O_3_-MnS-BN, and BN contributed to IGF formation. Zheng et al. [16] showed that the M_2_B precipitation phase can be formed because of the excessive B atoms segregating at the austenite grain boundaries, which not only promoted the hardenability during austenitization to deteriorate significantly but also obviously deteriorated toughness during tempering. Sun et al. [17] indicated that boron segregation can induce the appearance of cracks. Liu et al. [18] found that the increase in strengthening was attributed to the grain refining caused by the inhibition of boron on recrystallization with the increase in boron content. Liu et al. [19] demonstrated that the combinations of Ti and B elements can refine acicular ferrite and can decrease the number of prior austenite grain boundary (PAGB) ferrite in the weld metal and that B distributed at the PAGB can prevent the formation of PAGB ferrite.

Therefore, appropriate amounts of boron can effectively improve the impact toughness of steel, whereas excessive amounts of boron can cause the deterioration of impact toughness. In addition, the combination of boron and Ti can improve the impact toughness. However, the boron content in the weld metals has complex effects. Lee et al. [20] indicated that the impact toughness slightly decreased with the increase in boron contents from 32 to 60 ppm but significantly decreased with an increase in boron contents from 60 to 103 ppm. No cracks could be found when the boron content was in the range of 32 and 60 ppm; cracks were detected in the specimen when welded with 103 ppm boron. Kim et al. [21] found that boron segregation appeared in the welded joints and showed that boron segregation was also related to the welding heat input and to external stress. The higher heat input and external stress led to the decrease in the grain boundary segregation of boron. Da et al. [22] identified boron segregation at austenite grain boundaries by atom probe tomography (APT) and nano-secondary ion mass spectrometry (SIMS) technologies and also found that the boron segregation was directly related to the testing temperature. Minor boron can promote the formation of AF, can make the grain size finer, and can decrease the segregation of the S and the P, which can improve the impact toughness. However, unsuitable boron content leads to the formation of granular bainite and M-A constituents and the appearance of welding cracks. The suitable range of boron content in various steel types is different, so the study of boron content in 12Cr1MoVR low-alloy heat-resistant steel weld metals on the microstructure and impact toughness is necessary.

In this study, to reveal the effects of B contents on the impact toughness of the weld metals for 12Cr1MoVR low-alloy heat-resistant steel, three weld metals containing 0.0028%, 0.0054%, and 0.0079% B contents were designed. Next, the multipass weld metals were obtained. Finally, the microstructure, hardness, and the impact toughness were characterized and analyzed to investigate the effect of various B contents on the weld metals.

## 2. Materials and Procedure

In this study, 12Cr1MoVR low-alloy heat-resistant steel with a tensile strength of 660 MPa and a tensile elongation of 40% was chosen as the base metal (the chemical composition was listed in Table 1). The multi-pass weld metals used in this study were obtained by an automatic mixed gas shielded metal active gas (MAG) arc welding method with an OTCFD-V6 welding robot and a low-Cr heat-resistant steel metal powder-cored wires with three different B contents used as the filler metals. The diameter of the wires was 1.6 mm. The composition of the shielding gas was 80% Ar and 20% CO_2_, and the flow rate of the gas was 15–20 L/min. The welding heat input, welding current, welding voltage, and interpass temperature was 1.6 kJ/mm, 240 A, 29 V, and 100 °C, respectively. The content of the diffusible hydrogen was 4–4.5 mL/100g. The chemical compositions of the three weld metals are listed in Table 2, which were determined by an optical emission spectrometer (Spectro lab M9, SPECTRO, Kleve, Germany).

The purpose of the addition of B content was to increase the impact toughness of the weld metals. In our pre-testing experiments, a B content greater than 0.0079% led to the appearance of welding cracks. In this study, B contents in the range of 0.0028–0.0079% B were chosen. The schematic of the welding processes and the sampling schematic are shown in Figure 1. The specimens for microstructure observations of the weld metals were prepared from the welded joints. Charpy impact toughness specimens with the dimensions of 55 mm × 10 mm × 10 mm were prepared according to the GB/T 2650-2008 criterion [23]. The impact toughness experiments were performed at −40, −20, −10, 0, 20, 30, 60, and 80 °C. The hardness was measured by an HAT-1000A hardness tester at a load of 100 g and a loading time of 15 s according to the GB/T2654-2008 criterion. The hardness values of 10 points with the interval distance of 200 μm were measured and averaged as a final hardness result for each region. The metallographic samples with the dimensions of 30 mm × 15 mm × 10 mm were cut along the direction perpendicular to the welding length and were then ground, mechanically polished, and finally etched for 10–15 s with 2% HNO_3_ alcohol solution at room temperature. The microstructures of the weld metals and fracture surfaces of the impact specimens were observed and analyzed by scanning electron microscopy (SEM) (Quanta FEG 450, FEI, Hillsboro, OR, USA). The chemical compositions of the cleavage fracture initiation origin in Table 3, Table 4 and Table 5 were obtained by SEM with energy-dispersive X-ray spectroscopy (EDS, Oxford Instruments, London, United Kingdom). The sizes of inclusions were measured by SEM figures. The specimens for measuring austenite grain sizes were specially etched by a picric acid solution and were observed by SEM.

## 3. Experimental Results and Discussion

### 3.1. Effects of B Content on the Impact Toughness

Figure 2 presents the ductile–brittle transition curve between the impact toughness and testing temperature. In Figure 2, the average value and error bar of the impact toughness at each testing temperature is shown. In this study, the temperature, which corresponds to the average value of the impact energy at −40 °C of the bottom-shelf zone and the impact energy at 80 °C of the top-shelf zone, is regarded as the ductile–brittle transition temperature (DBTT). DBTT is 30, 0, and −14 °C, for B1, B2, and B3, respectively. We found that the DBTT decreases and the impact toughness increases with the increase in B content from 0.0028% to 0.0079%. However, at the top-shelf zone, the inverse change trend was found. The average value of the impact energy of the B1, B2, and B3 specimens at the testing temperature of 80 °C were 179, 152, and 135 J, respectively. The change in DBTT and in the top-shelf zone with the different B contents was analyzed using the microstructure, the prior austenite grain size, the inclusion, the fracture surface, and the hardness.

### 3.2. Effects of B Content on the Microstructure and the Prior Austenite Grain Size

Figure 3 shows the macro-features of the three weld metals B1, B2, and B3 with the three B contents. From Figure 3, the weld metals are all composed of a columnar grain zone (CGZ) and a reheated zone (RZ). The microstructures of the columnar grain zone and the reheated grain zone are shown in Figure 4. By comparison, it was found that the microstructure of the columnar grain zone and the reheated zone were all composed of the block ferrite and minor inclusions and carbides. Figure 5 shows the feature of the prior austenite grain in the columnar grain zone of the weld metals with various B contents. As shown in Figure 5, the prior austenite grain size decreases with the increase in B contents. The sizes of the prior austenite grain were measured as shown in Figure 6. From Figure 6, the sizes of the prior austenite grain of the B1, B2, B3 weld metals were in the range of 60–200, 40–140, and 40–120 μm, respectively. A large number of studies have shown that the addition of boron usually causes the redistribution of grain boundary energy caused by boron segregation [15], reduces the number of ferrite heterogeneity nucleation [7], causes boron segregation on the prior austenite grain boundary [23,24], and delays the transformation process from austenite to ferrite during cooling [21,25]. In our study, although the distribution of B could not be identified, the refinement of grain size could be found because of the addition of B content. As shown in Figure 5 and Figure 6, with the increase in boron content, the size of the prior austenite grain in the weld metal decreases, which hinders the crack propagation and improves the toughness of the weld metal. The change of the prior austenite grain size can explain the increase in the impact toughness with the increase in B content. However, for the top-shelf zone, the change in the impact toughness was analyzed by the inclusions.

### 3.3. Effects of B Content on Hardness 

The hardness of the weld metals is directly related to the microstructure and the grain sizes. To demonstrate the effects of B contents on hardness, the hardness of the columnar grain zone and the reheated zone were measured. Figure 7 shows the hardness distribution of the three weld metals. The average hardness values of the columnar grain zone for the three weld metals B1, B2, and B3 were 231, 222, and 224 HV, respectively. The average hardness values of the reheated zone for the three weld metals B1, B2, and B3 were 220, 203, and 205 HV, respectively. As shown in Figure 7, we found that minor differences exist in the three weld metals. For each weld metal, the hardness of the reheated zone was lower than that of the columnar grain zone. This was caused by the grain refinement of the reheated zone, which was induced by the decrease in the grain size. In Figure 6, the prior austenite grain size decreases with the increase in B contents from 0.0028% to 0.0054% to 0.0079%. Based on the theory that fine grain makes the materials stronger and harder, the hardness of the finer reheated zone is higher than that of the coarser columnar grain zone. This is consistent with Liu et al. [18]. Ref. [18] reported that an increase in strength was attributed to the grain refining caused by the inhibition of boron on recrystallization with the increase in boron content.

### 3.4. Effects of B Content on Impact Fracture Surface

Figure 8 shows the fracture surface of the B1 weld metal at 60 °C. Three main micro-parameters including stretch zone width (SZW), fibrous crack length (SCL), and the distance (X_f_) from the crack initiation site to the fibrous crack were marked in the impact fracture surface in Figure 8. The relationships between micro-parameters and the impact toughness of the weld metals with the three B contents were redrawn in Figure 9. In Figure 9, the impact toughness increases with the increase in SZW + SCL. The value of SZW + SCL directly reflects the value of the impact toughness. In Figure 8, the specimen is fractured at the end of the cleavage fracture. The early or late of the cleavage fracture directly determines the level of the impact toughness (whether lower or higher). The longer the SZW + SCL is, the later the cleavage fracture is; thus, a higher impact toughness can be obtained. Additional theory is referred to in ref. [26].

To reveal the differences of the DBTT curves of the three weld metals, the fracture surfaces of the three typical regions including the bottom-shelf zone, the ductile–brittle transition zone, and the top-shelf zone were carefully analyzed. Figure 10 shows the fracture surface of the three specimens with various B contents at −40 °C in the bottom-shelf zone of Figure 2. At −40 °C, cleavage fracture features dominated the fracture surfaces of all specimens, and with the increase in B content, the fraction of the cleavage fracture zone slightly decreased, whereas the fractions of the dimple fracture zone and the shear fracture zone increased. In the cleavage fracture zone, the final cleavage fracture is initiated at the inclusions, which is identified as the oxide-inclusion containing Fe, Mn, Ti, Al, and Si (shown in Table 3, Table 4 and Table 5). In the bottom-shelf zone, grain size becomes the dominant factor affecting the impact toughness. The coarser grain size in the specimen with 0.0028% B forms the larger cleavage fracture facets, which induces the lower impact toughness (which is consistent with ref. [27]).

Figure 11 shows the fracture surface of the three specimens with various B contents at 0 °C in the ductile–brittle transition zone of Figure 2. At 0 °C, quasi-cleavage fracture features dominated the fracture surfaces of all specimens, and with the increase in B content, the cleavage fracture zone fraction decreased, whereas the fractions of the dimple fracture zone and the shear fracture zone increased; this means the impact value at the same testing temperature increased with the increase in B content. In the cleavage fracture zone, the final cleavage fracture was initiated at the inclusions, which is identified as the oxide-inclusion containing Fe, Mn, Ti, Al, and Si.

Figure 12 shows the fracture surface of the three specimens with various B contents at 80 °C in the top-shelf zone of Figure 2. At 80 °C, the ductile fracture dominates the fracture surface. On the fracture surface, there are no cleavage fracture zones, but only a dimple fracture zone and a shear fracture zone. Several studies [28,29] have demonstrated that the inclusion-induced void formation, void coalescence, dimple formation, dimple growth, and the final fracture are the main stages of dimple fracture. From Figure 12d,e,f the inclusion-induced dimple fracture feature controlled the fracture surface, and more inclusions could be found in the B1 weld metal. The dimple size of the fracture surface of specimen B1 was larger and the hole was deeper, whereas the dimple shape of B2 and B3 gradually became shallow and the size became smaller. When the fracture conditions are the same, the larger the dimple size, the deeper and more uniform the dimple is; additionally, more energy is consumed when ductile fracture occurs, so the impact absorption energy of the B1 specimen is larger and the toughness of the B1 specimen is better.

### 3.5. Effects of B Content on the Inclusions

To further investigate the effects of different boron contents on the impact toughness of weld metals in the top-shelf zone, it is necessary to study the effect of boron on inclusions in the weld metals. In Figure 4, the inclusions are distributed near the grain boundary. In this study, the chemical composition of the three weld metals B1, B2, and B3 was roughly the same and the dimple formation stages of the three weld metals were similar, so the size and density of the inclusions in the weld metals were the main factors affecting the impact properties. To further analyze impact toughness at the top-shelf zone, the size and density of the inclusions in the weld metals with the three different boron contents at 80 °C were measured. Figure 13 shows the statistics of the inclusions in the weld metals with the different boron contents on the impact fracture surface of the weld metal at 80 °C. Here, the diameter of inclusion is measured as the size of the inclusion, and the area percentage of the inclusion area in the selected area is regarded as the inclusion density. As shown in Figure 13a, we found that the average sizes of the inclusions were 0.54, 0.61, and 0.66 μm, which increased slightly with the increase in B contents. The highest proportion was in the range of 0.5–1 μm. The inclusions with large sizes appeared in the B3 specimens. In Figure 13b, with the increase in boron contents, the area ratios of inclusions increased from 0.2% to 0.9%. Dimple fracture is mainly caused by the expansion and convergence of the cavities formed by the separation of inclusions and interfaces between grains. When the applied load acts on the impact specimen, the dimples formed by the inclusions with smaller size are smaller and the distance between them is longer, so more energy is consumed. However, the cluster inclusions with large size and high density destroy the microstructure continuity of the weld metals, form large cavities, and hinder the mutual convergence distance of inclusions; so, it requires less energy consumption and leads to poor toughness. With the increase in boron content, the size and density of inclusions under the impact fracture at 80 °C increase gradually, and the energy required for ductile fracture under the action of external force is lowered. Therefore, the toughness anomaly occurs at 80 °C, that is, for the top-shelf, and the impact toughness is lower with the increase in B content.

## 4. Conclusions

In this study, the weld metals of 12Cr1MoVR low-alloy heat-resistant steel was obtained by automatic mixed gas shielded MAG welding methods and the effects of B contents on the microstructure and impact toughness of heat-resistant steel weld metals were investigated. The hardness and impact toughness of three kinds of weld metals with different boron contents (B1 = 0.0028%, B2 = 0.0054%, and B3 = 0.0079%) were measured, and the macro- and micro-morphology of impact fractures were observed. The microstructure changes of the weld metals with different boron contents and the size and density changes of inclusions in the weld metals with three different boron contents at 80 °C were analyzed by SEM. The following conclusions were obtained:

(1) The increase in boron content cannot change the microstructure type of the weld metals but can make the prior austenite grain size decrease.

(2) With the increase in boron content, the ductile–brittle transition temperature of the weld metals decreases and the low temperature impact toughness of the weld metals increases, which can be explained by a decrease in the prior austenite grain size with the increase in boron content. The finer prior austenite grain sizes can hinder the crack propagation and can improve the low temperature impact toughness of the weld metals.

(3) For the top-shelf zone, the change in the impact toughness is caused by the inclusions. When the testing temperature was 80 °C, the dimple fracture dominated the fracture mode. With the increase in B content, the size and density of inclusions increased gradually, which makes convergence and formation of the dimples easier. Therefore, the impact toughness was the lowest when the B content was 0.0079% at 80 °C.

(4) To clearly achieve high-precision qualitative and quantitative analysis about the distribution and existent forms of B, the combined characterization of atom probe tomography and secondary-ion mass spectrometry should be performed.

## Figures and Tables

**Figure 1 materials-14-00926-f001:**
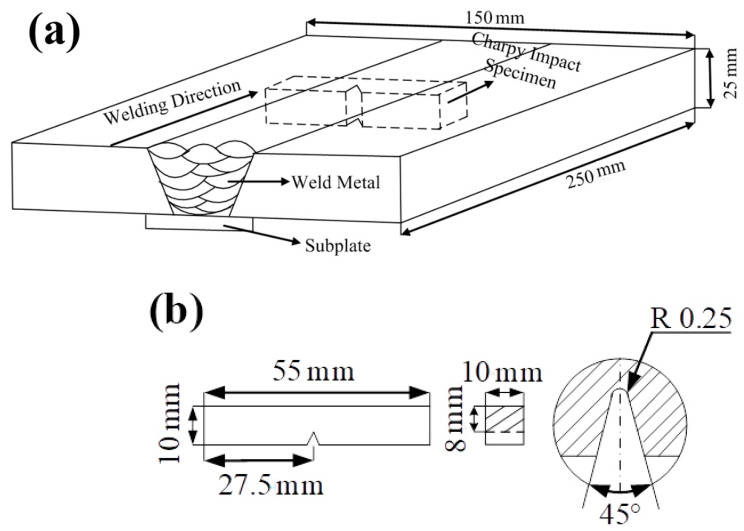
(**a**) Schematic illustration of the welding processes and the Charpy impact toughness specimen cut from the multi-layer, multi-pass weld metal; (**b**) dimensions of the Charpy impact toughness specimen.

**Figure 2 materials-14-00926-f002:**
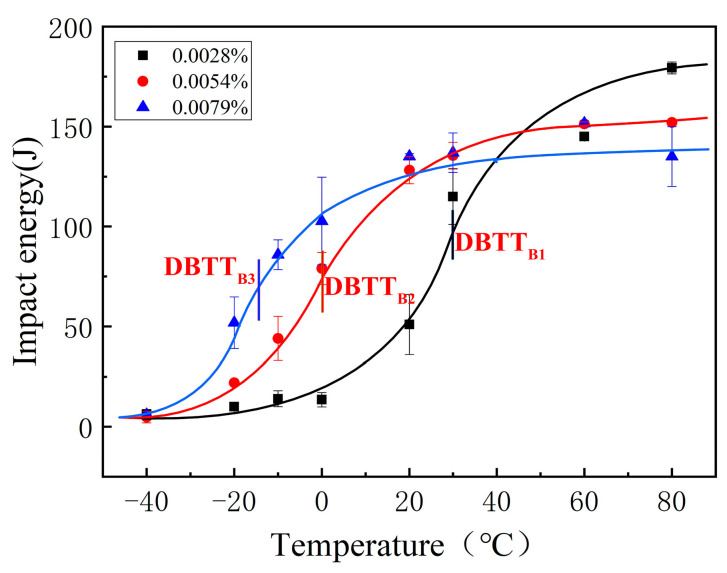
Ductile–brittle transition curve of the three weld metals with 0.0028% B (B1), 0.0054% B (B2), and 0.0079% B (B3).

**Figure 3 materials-14-00926-f003:**
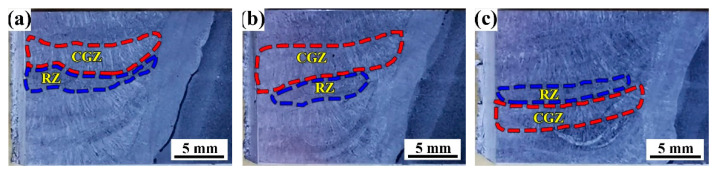
Macro-features of the three weld metals. (**a**) 0.0028 % B (B1); (**b**) 0.0054 % B (B2); (**c**) 0.0079 % B (B3).

**Figure 4 materials-14-00926-f004:**
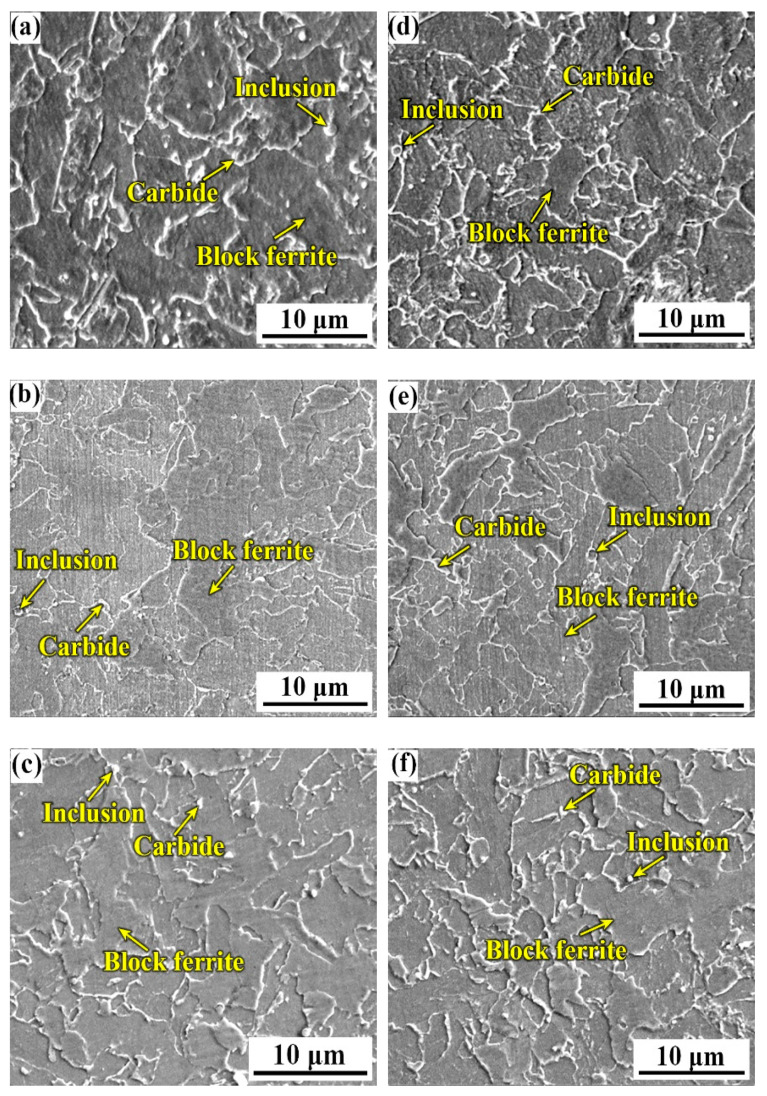
Microstructures of the three weld metals. (**a**) Columnar grain zone-0.0028% B (B1); (**b**) columnar grain zone-0.0054% B (B2); (**c**) columnar grain zone-0.0079% B (B3); (**d**) reheated zone-0.0028% B (B1); (**e**) reheated zone-0.0054% B (B2); (**f**) reheated zone-0.0079% B (B3).

**Figure 5 materials-14-00926-f005:**
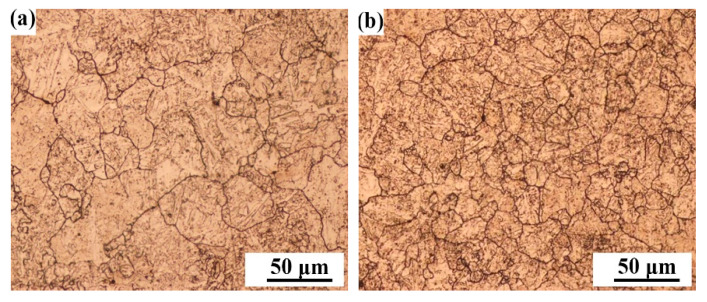
Prior austenite grain boundaries of the three weld metals. (**a**) 0.0028% B (B1); (**b**) 0.0054% B (B2); (**c**) 0.0079% B (B3).

**Figure 6 materials-14-00926-f006:**
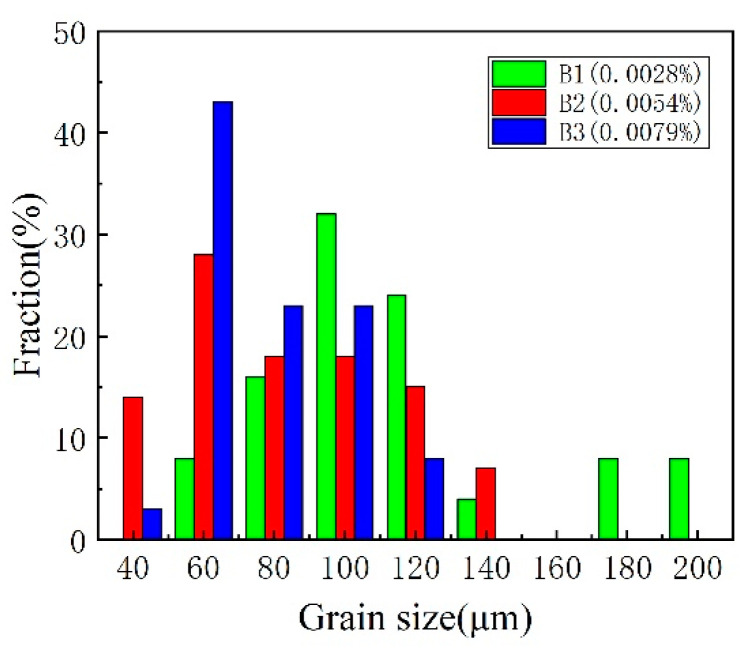
Distribution of prior austenite grain size in the weld metals with three different boron contents.

**Figure 7 materials-14-00926-f007:**
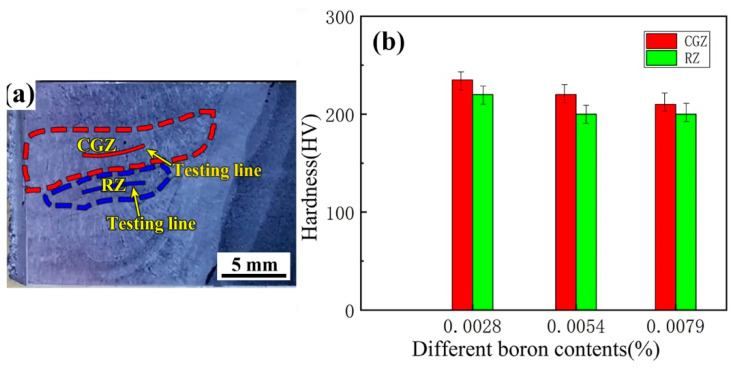
Hardness of the three weld metals with 0.0028% B (B1), 0.0054% B (B2), and 0.0079% B (B3). (**a**) Schematic measuring hardness; (**b**) relationships between hardness and boron contents.

**Figure 8 materials-14-00926-f008:**
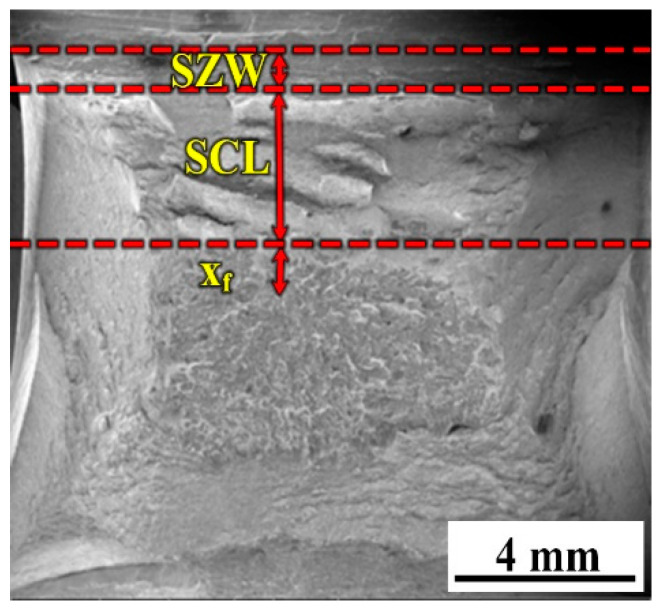
Three micro-parameters on a typical fracture surface of the weld metal with 0.0028% B (B1) at 60 °C.

**Figure 9 materials-14-00926-f009:**
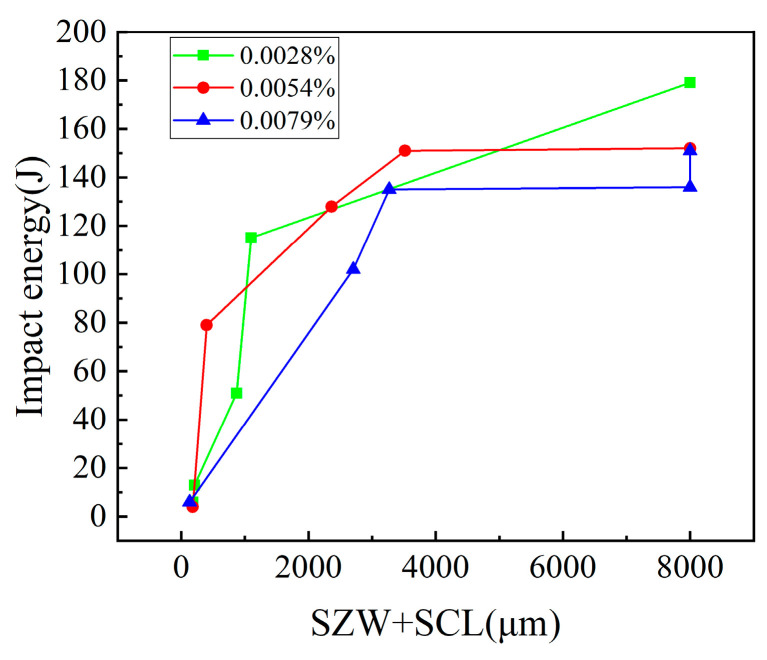
Relationships between impact toughness and SZW + SCL.

**Figure 10 materials-14-00926-f010:**
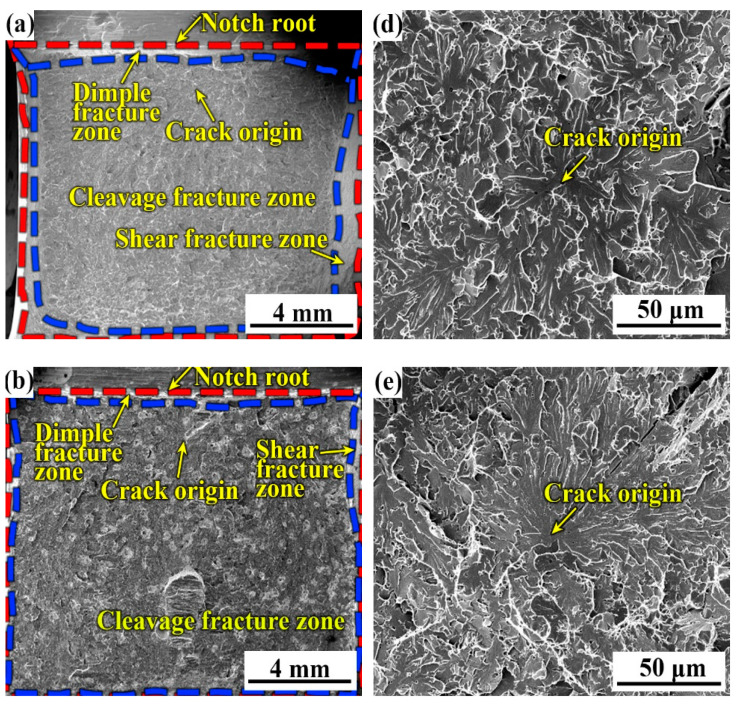
Typical fracture surfaces of three weld metals at −40 °C. (**a**,**d**) B1; (**b**,**e**) B2; (**c**,**f**) B3.

**Figure 11 materials-14-00926-f011:**
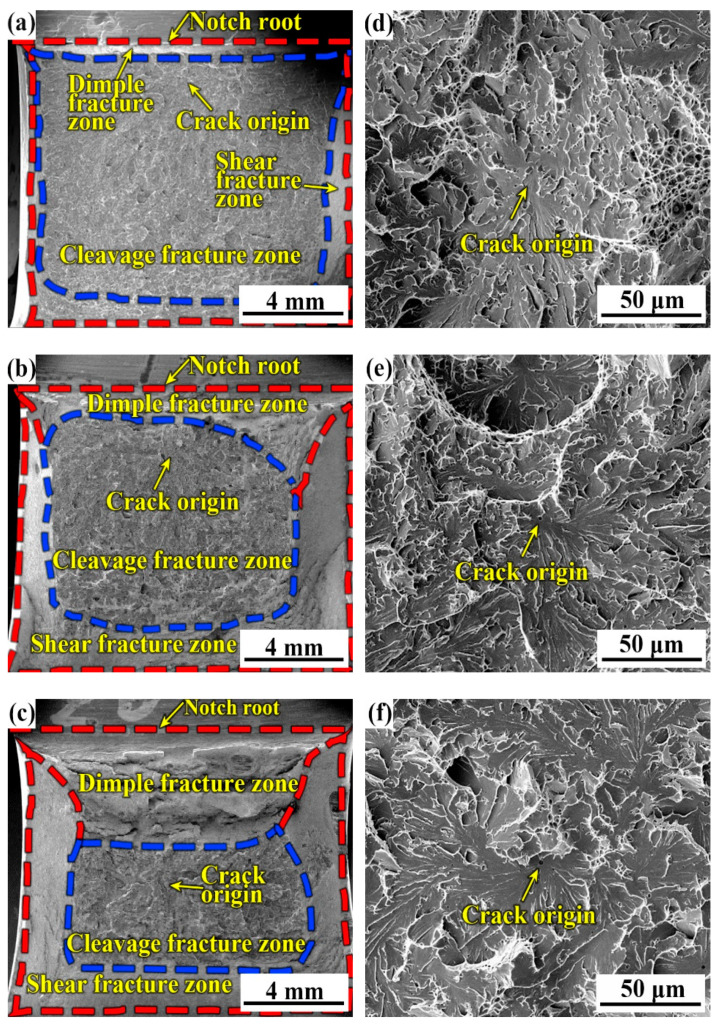
Typical fracture surfaces of three weld metals at 0 °C. (**a**,**d**) B1; (**b**,**e**) B2; (**c**,**f**) B3.

**Figure 12 materials-14-00926-f012:**
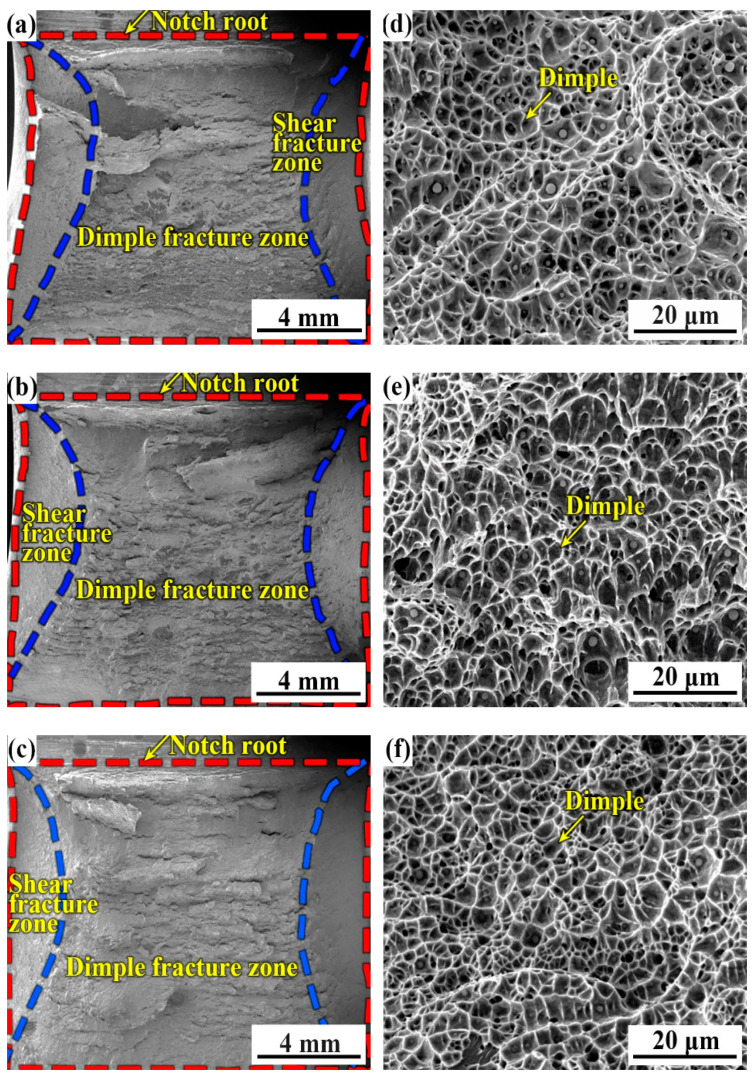
Typical fracture surfaces of the three weld metals at 80 °C. (**a**,**d**) B1; (**b**,**e**) B2; (**c**,**f**) B3.

**Figure 13 materials-14-00926-f013:**
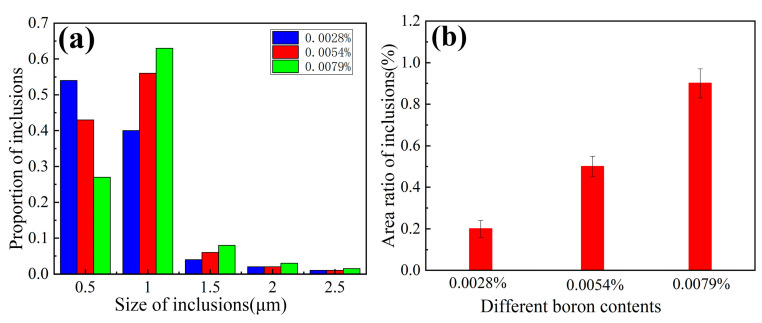
Inclusions of the weld metals with various B contents. (**a**) Proportion of inclusions; (**b**) area ratio of inclusions.

**Table 1 materials-14-00926-t001:** The main chemical composition of 12Cr1MoVR low-alloy heat-resistant steel (%).

Elements	C	Si	Mn	Cr	Mo	V	Others
12Cr1MoVR	0.1	0.3	0.7	0.90	0.35	0.15	0.5

**Table 2 materials-14-00926-t002:** The main chemical composition of the three weld-deposited metals (%).

Types/Elements	C	Mn	Si	Cr	Mo	V	B	Others
B1	0.045	0.98	0.30	0.90	0.42	0.15	0.0028	0.40
B2	0.041	1.07	0.32	0.94	0.42	0.15	0.0054	0.43
B3	0.041	1.04	0.33	0.89	0.41	0.15	0.0079	0.54

**Table 3 materials-14-00926-t003:** The main chemical composition of the initiation origin in the fracture surface of the B1 weld metal (%).

Temperature/Elements	Fe	O	Mn	Ti	Al	Si
−40 °C	40.8	34.9	7.2	7.6	4.5	5.0
0 °C	77.4	13.8	0.8	6.4	1.0	0.6

**Table 4 materials-14-00926-t004:** The main chemical composition of the initiation origin in the fracture surface of the B2 weld metal (%).

Temperature/Elements	Fe	O	Mn	Ti	Al	Si
−40 °C	45.4	36.1	16.4	10.7	3.6	2.8
0 °C	36.4	27.5	15	12.1	5.7	3.3

**Table 5 materials-14-00926-t005:** The main chemical composition of the initiation origin in the fracture surface of the B3 weld metal (%).

Temperature/Elements	Fe	O	Mn	Ti	Al	Si
−40 °C	40.5	37.3	7.4	8.3	2.4	4.1
0 °C	25.5	43.8	11.6	14.0	3.8	1.3

## Data Availability

Data is contained within the article or supplementary material. The data presented in this study are available in the form of the final report from the project TECHMATSTRATEG. The report is open only for the authors of the article.

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
