# Peer review of "Effects of Boron Content on the Microstructure and Impact Toughness of 12Cr1MoVR Low-Alloy Heat-Resistant Steel Weld Metals"

_materials, 2021, doi:10.3390/ma14040926_

Round 1
Reviewer 1 Report
Dear Authors,
I have reviewed your paper "Effects of Boron content on the microstructure and impact toughness of heat resistance steel weld metals".
The paper presents interesting results, and could be considered for publishing after some improvements.
General remarks:
- You have presented 20 references, in which only 4 have been published in last three years. The science made big step forward in this period, so it is hard to believe, no more articles are relevant to show scientific background or discussion.
- Please add spaces before brackets. Also, the spaces inside brackets should be deleted, e.g., [1, 2].
- The propper phrase is "et al." not "et al".
Introduction:
- Line 66 – add space before bracket.
- Lines 72, 83, 85, 91, 96, 99, 100 – the same as above.
- Lines 108-109 – what new is in your paper? You should clearly mark the novelty of your work. Mark new aims and ideas in your investigations.
Materials and procedure:
- Move “p” to “P” in the name of the paragraph. It should be “Materials and Procedure”.
- Line 115, are you sure, that the unit is “kJ/cm”? Or “kJ/mm”? The proper unit of heat input (following many welding standards) is “kJ/mm”. Why have you used presented (line 115) values of welding parameters? The description is missing. Why parameters for each welding pass were the same? How many
- If it is possible, please clearly mark the grades of used steels. Also, their mechanical properties (Re, Rm, A) should be presented.
- The information about filler materials are missing – chemical composition and mechanical properties.
- Have you used any requirements from relevant standards (impact test, hardness measurements, metallographic test) in your research? If yes, please add their numbers. If not, please mark, why your investigations were not performed following welding engineering standards.
Experimental results and Discussion:
- Move “r” to “R” in the name of the paragraph. It should be “Experimental and Results and Discussion”.
- “3.1.” and “3.3.” – I propose to add table with results. It will be more clear.
- I cannot find any scientific discussion in subsections 3.3., 3.4., and 3.5.. Please compare your results with other scientists. You should clearly mark the advantages of your investigations.
Summary:
- I propose change the name of sections to “Conclusions”.
- Section is strongly connected with the paper, I have no comment here.
Reviewer 2 Report
An adjustment is required in part:
1) The title of the article should reflect the novelty. Therefore, it is necessary to add the type of steel, i.e. steel with a low chromium content;
2) The section "Materials and procedure" should be renamed to the section "Materials and methods" and expanded: specify the method for determining the chemical composition of the weld, the method for determining the chemical composition of oxide inclusions (Tables 2, 3 and 4), the method for determining the size of inclusions (line 275), as well as the method for etching samples for 10...15 seconds to identify the austenite grain. For the validity of the research, they must be added;
3) In Table 1, eliminate typos: in option B1 for Si and Cr, add digits, and in option B3 for Si, remove the extra dots;
4) In Figure 6, you need to correct the error: specify the shares of the unit or the scale to represent as a percentage;
5) In the manuscript, it is necessary to explain how such a slight change in the grain hardness from 5 to 12% affects the properties of steel (see Figure 7);
6) More clearly state the physical meaning of the graphic dependencies of Figure 9;
7) To complete the analysis of the experimental dependence of impact strength on temperature at different content of boron in the style (figure 2) must be supplemented reasoning according to another characteristic of the region, namely the "convergence" of the graphs at a temperature of + 40C.
Reviewer 3 Report
In the paper are presented a series of information on the effect of boron content on the characteristics of welded steel joints.
From the analysis of the information presented in the article, I found the following:
- The paper presents a series of results that could be of interest to the scientific community;
- the title of the paper should be changed as it highlights only part of the research without highlighting the hardness tests and the tendency to form inclusions;
- the introduction needs to be improved by taking into account other newer bibliographic sources. At the end of the introduction, the detailed structure of the paper and the research objectives must be presented.
- the research methodology should be presented in more detail as it is not very clear. Thus, it should be explained why a change in the boron composition was considered only in the range of 0.0028-0.0079%.
- the chemical composition of the wires used for welding and the manufacturer must be clearly specified;
- it is necessary to provide details on the number of weld layers deposited and whether they were all made with the same welding regime;
- it must be specified whether the chemical composition of the steels, shown in Table 1, was determined by the authors or was taken from a specific bibliographic source;
- it is necessary to present macroscopic images of the welded samples;
- it is necessary to present a sketch of the points on the samples where the hardness measurement was performed;
- for the chemical compositions presented in Tables 2, 3 and 4 respectively it is necessary to present the method by which the determination of the chemical composition and the equipment used was performed;
- the experimental results section should not start without a small introduction;
- in the discussion part, the causes that determined the obtaining of those results in research must be better explained. A quantitative presentation of the results alone is not enough. The novelty of the research carried out in relation to other research in the field must also be highlighted very clearly;
- in the final part of the conclusions, the future research directions and the possible practical applications of these materials must be presented.
Round 2
Reviewer 1 Report
Dear Authors,
Your efforts are appritiate. The paper has been improved a lot. My overal merit about your work is good. However, it still needs improvement.
Remarks:
- Following the EN 1011-1:2009 "Welding - Recommendations for welding of metallic materials - Part 1: General guidance for arc welding" the unit for heat input should be "kJ/mm".
The same unit "kJ/mm" is marked in ISO/TR 18491:2015 "Welding and allied processes — Guidelines for measurement of welding energies".
I propose to present heat input as "kJ/mm", not "kJ/cm". - "The information about filler materials are missing –chemical composition and mechanical properties.
Response: Because the factors affecting the microstructure and impact toughness of the weld metals is the chemical compositions of the weld metals, so the chemical compositions of the weld metals have been presented."
I cannot agree with this statement, that microstructure depends only by chemical composition of weld metals. The filler material also affect the microstructure of welded joint - e.g., the effect of diffusible hydrogen, which can diffuse to the joint strongly affects the metallurgical aspects, and the susceptibility to cold cracking. Also, the inclusions from filler material affects the microstructure of the joint. There are some investigations, showed e.g., how waterproof coatings laid on the surface of filler materia, affect the quality and properties of welded joints. The usage of the same filler material, covered only by additional substances, provide much different properties and microstructures.
Following this reason, I still recommend to add the chemical composition and properties of filler material (deposited metal). - “3.1.” and “3.3.” –I propose to add table with results. It will be more clear.
Response: I think that the figures are clearer than Tables. For Impact toughness, more scholars used the ductile-brittle transition curve to demonstrate the difference of the impact toughness. If tables were used, impact toughness of three specimens has been measured at each testing temperature, data will very big."
I fully agree that Figures are clearer. However, in your figures I cannot find values near the presented points, e.g. Fig. 2. In the text (line 168) you have presented only three values, and in the picture, there are more points.
If you do not want to present tables (which I understand due to very big data), please add values in the picture.
Best regards,
Reviewer
Reviewer 2 Report
Remarks worked well. There is no question.
Reviewer 3 Report
The authors revised their manuscript according to my suggestions. Thus the manuscript can be accepted for publication.
